# Concept Protocol for Developing a DAid® Smart Socks-Based Biofeedback System: Enhancing Injury Prevention in Football Through Real-Time Biomechanical Monitoring and Mixed Reality Feedback

**Anna Davidovica** [1,*]📷, **Guna Semjonova** [1], **Lydia Kamynina** [2], **Linda Lancere** [3], **Alise Jonate** [1], **Signe Tomsone** [1]📷, **Aleksejs Katasevs** [4], **Aleksandrs Okss** [5] **and Sergejs Davidovics** [1]

1    Department of Rehabilitation, Riga Stradins University, 16 Dzirciema Street, LV-1007 Riga, Latvia; guna.semjonova@rsu.lv (G.S.); alise.jonate@gmail.com (A.J.); signe.tomsone@rsu.lv (S.T.); sergejs.davidovics@gmail.com (S.D.)

2    Department of Engineering, Vidzeme University of Applied Sciences, LV-4201 Valmiera, Latvia; lydia.kamynina@va.lv

3    Department of Sociotechnical Systems Modelling, Vidzeme University of Applied Sciences, LV-4201 Valmiera, Latvia; linda.lancere@gmail.com

4    Institute of Mechanical and Biomedical Engineering, Riga Technical University, LV-1048 Riga, Latvia; aleksejs.katasevs@rtu.lv

5    Institute of Architecture and Design, Riga Technical University, LV-1048 Riga, Latvia; aleksandrs.okss@rtu.lv

\*    Correspondence: anna.davidovica@rsu.lv

**Abstract:** Football players, particularly in youth leagues, face a high risk of lower limb injuries due to improper movement patterns. While programs like FIFA 11+ help reduce injuries, they lack real-time, personalized feedback for biomechanical correction. This concept protocol outlines the development of a DAid® smart socks-based biofeedback system that integrates biomechanical monitoring with mixed reality (MR) feedback to enhance injury prevention. The DAid® smart socks, equipped with pressure sensors and inertial measurement units (IMUs), track plantar pressure distribution and the center of pressure (COP). Real-time feedback is delivered via a Meta Quest 3 MR headset, enabling athletes to adjust movement patterns instantly. This protocol establishes a framework for evaluating the system's feasibility and effectiveness in optimizing biomechanics and reducing injury risks. By combining wearable technology with MR-based feedback, this study advances injury prevention strategies, with potential applications in rehabilitation and performance training.

**Keywords:** smart textiles; biofeedback; football injury prevention; biomechanics; mixed reality



## 1. Introduction

Football is one of the most physically demanding sports, with players frequently exposed to a considerable risk of injuries, particularly in the lower extremities [1]. These injuries not only affect individual performance but also have significant implications for team dynamics and overall game outcomes [2]. Common lower limb injuries in football often result from improper movement patterns or biomechanical errors, such as poor ankle stability, or asymmetrical weight distribution, like dynamic knee valgus, where the knee collapses inward during activity [3,4]. These biomechanical errors place excessive stress on joints and soft tissues, increasing the likelihood of injuries like anterior cruciate

ligament (ACL) tears, ankle sprains, or muscle strains [5]. Biomechanics plays a crucial role in injury prevention by identifying movement inefficiencies and correcting faulty mechanics before they lead to injuries [6], especially during high-impact movements—such as rapid accelerations, decelerations, cutting maneuvers, and landings—are particularly susceptible to these biomechanical errors, making real-time monitoring and correction essential for injury prevention [7]. Several key biomechanical markers indicate injury risks, including joint angles, plantar pressure distribution, and the center of pressure (COP). These parameters provide insights into an athlete's movement quality and highlight areas requiring intervention. For example, an improper COP shift during landing can signal a high risk for ACL injuries, while uneven plantar pressure distribution may contribute to chronic overuse injuries. Identifying these risk factors enables targeted interventions, such as neuromuscular training and movement re-education, to optimize movement efficiency and reduce injury susceptibility [8]. In response to these challenges, injury prevention programs like FIFA 11+ have been developed to improve players' strength, balance, and neuromuscular control. The FIFA 11+ program, developed by the Fédération Internationale de Football Association (FIFA), is a comprehensive injury prevention initiative for football players [9,10]. It consists of warm-up routines, strength training, balance exercises, and agility drills aimed at reducing injury risk, especially for the lower limbs [11]. Despite its success in preventing injuries [9,10], it lacks personalized, real-time feedback for monitoring and correcting players' movements, leading athletes to perform exercises incorrectly, reducing players' effectiveness and leaving them susceptible to injury [12].

Various devices have been introduced to monitor movement quality [13]. These include wearable sensors and integrated systems within training equipment that offer real-time feedback on performance, allowing for immediate adjustments [14]. Despite these advancements, many current devices suffer from issues such as rigidity [15], discomfort [16], and sensor displacement [17] leading to inaccurate data and limited user acceptance [18,19].

Smart textiles offer a promising solution to these challenges [20]. Smart textiles, or e-textiles, integrate electronic components directly into fabrics, creating flexible, wearable devices that conform to the body's movements [20,21]. For instance, smart socks equipped with pressure sensors and inertial measurement units (IMUs) can measure critical parameters such as plantar pressure distribution and ankle joint positioning [22,23]. These smart socks provide continuous biomechanical monitoring without the discomfort associated with traditional devices [24]. However, applying this technology specifically to youth football players presents new challenges, including adapting the approach to the rigorous training demands of young athletes. Our interdisciplinary research project aims to address these challenges by developing through conceptualization a biofeedback method based on DAid® smart socks for youth football players. This biofeedback system will measure key movement quality biomechanical parameters in real-time, including pressure values under each foot's plantar area and the center of pressure (COP) coordinates. The data will be used to provide immediate feedback on movement patterns, allowing athletes to correct improper techniques that could lead to injuries. This technology promises to enhance the effectiveness of injury prevention programs like FIFA 11+ and offer a more personalized approach to training.

Figure 1 provides an overview of the proposed biofeedback system, illustrating how DAid® smart socks will be used to measure key biomechanical parameters and deliver real-time feedback through a mixed reality interface.

The main contributions and rationales of our study are as follows:

- We present the development of a DAid® smart socks-based biofeedback system designing process combining methods from smart textile design, healthcare, and data analytics in developing a biofeedback method for youth football league players. This

rationale is based on the need to develop a user-friendly biofeedback method that highlights technology practical applications in sports injury prevention training.

- Our study is a concept protocol for the smart textile biofeedback method for injury prevention in the football youth league players population. The feedback will be delivered through a mixed reality (MR) head-mounted display (HMD), enabling real-time, immersive guidance on exercise performance. The rationale is to address the current gap in effective injury prevention methods by incorporating mixed reality (MR) feedback, thus providing real-time guidance that traditional methods might lack.

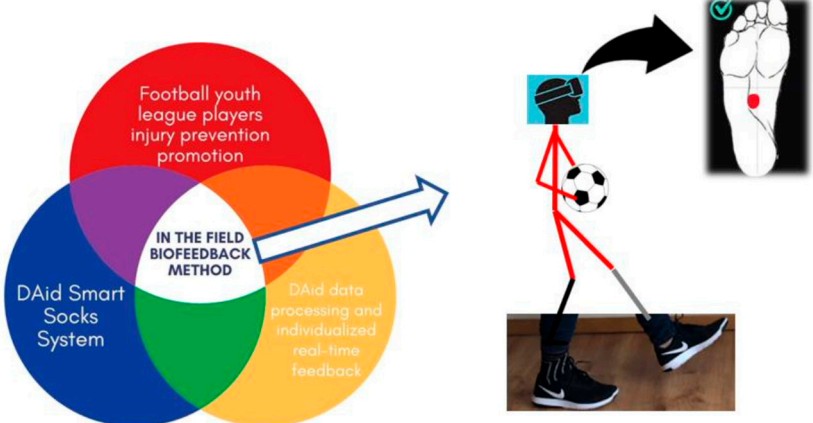

**Figure 1.** Conceptual framework of the smart socks-based biofeedback system.

## 2. Background and Related Work

### 2.1. Wireless Smart Sensors in Movement Monitoring

Advanced wireless smart sensor systems are playing a key role in monitoring lower limb movement and biomechanical parameters, offering real-time biofeedback that can optimize performance and prevent injuries [25]. Inertial measurement units (IMUs), which measure angular rate, magnetic field, and acceleration, provide critical data on movement dynamics that help in understanding biomechanical performance and adjusting training [26,27]. Electromyographic (EMG) sensors, on the other hand, track muscle activity, enabling the analysis of muscle fatigue, activation patterns, and overall muscle health, which is crucial for optimizing training [28,29]. For lower extremity monitoring, smart insoles like the Pedar and PODOSmart® systems capture pressure distribution and the center of pressure (CoP), offering detailed insights into foot biomechanics [30,31]. Similarly, smart socks like DAid® integrate piezoresistive textile sensors, which improve comfort while providing accurate biomechanical data. These systems are advantageous in sports medicine due to their affordability and ease of integration into training protocols, despite not offering the precision of laboratory-grade equipment [20,22,30,32,33]. However, these sensor systems face some challenges. When multiple wireless devices are used simultaneously, synchronizing data flow can become complex and time-consuming. Signal overlap and inaccuracies may occur and managing device activation timing adds to the complexity [34]. DAid® smart socks stand out by offering an unobtrusive design with a minimal impact on movement, making them a promising tool for more natural and effective monitoring [20,22,23,35]. Despite these advances, current smart textile technologies have yet to address the gap in biofeedback systems for injury prevention, particularly for youth football players. Addressing this gap could significantly enhance the effectiveness of movement correctness monitoring and injury prevention strategies for, e.g., by utilizing virtual reality-based solutions. Integrating real-time biofeedback from technologies like DAid® smart socks into the injury prevention programs like the FIFA 11+ program in a VR environment offers an opportunity to address these challenges. By using the smart socks

to monitor plantar pressure and the CoP, players can receive immediate visual feedback through a head-mounted display (HMD) if their foot alignment or weight distribution is off. For example, during exercises like the single-leg stance or squats, a red arrow could appear if there is an inward foot roll, prompting the player to adjust their posture [36]. This integration of biofeedback into injury prevention exercises can significantly improve movement precision, ensuring that players make the necessary adjustments to prevent injury.

*2.2. Virtual Reality Applications in Football*

Virtual reality (VR) is becoming a valuable tool in football training, especially in injury prevention and performance enhancement through biomechanical feedback and mixed reality simulations. While VR has been successfully applied in neurorehabilitation for stroke patients [37,38], in football, it is primarily used to simulate real-life game scenarios, helping players improve their tactical awareness, decision-making, and physical response to various in-game situations [39]. By training players to internalize safer movement patterns in a controlled environment, VR can reduce the risk of injury caused by poor technique or bad decision-making under pressure [37,40]. Additionally, VR has been used to enhance goalkeeping skills, allowing players to practice reflexes and decision-making without the physical risks of traditional on-field training [39]. VR also aids in technique improvement by offering a controlled space for athletes to refine their skills, reducing the likelihood of injury during actual play [37,41]. For injured players, VR can support rehabilitation by helping them maintain cognitive sharpness and engage in biomechanical analysis and proprioceptive exercises, contributing to effective recovery and injury prevention [41]. Leading VR providers [39,42] like ViPER, STRIVR, and REZZIL have developed specialized programs that include eye-movement tracking, heart rate monitoring, and biomechanical feedback to enhance player performance [41]. Despite its success, VR in football training has not fully addressed the specific challenge of monitoring and correcting movement patterns during injury prevention exercises for youth players. While VR tools improve decision-making and cognitive skills, they do not yet provide real-time biomechanical feedback in a way that is tailored to injury prevention, particularly in youth football contexts. Therefore, we present our conceptualization process for developing a DAid® smart socks-based biofeedback system to enhance injury prevention in football through real-time biomechanical monitoring and mixed reality feedback.

## 3. Design Process

The design process builds upon previous findings from testing the DAid® smart socks system (Riga Technical University, Riga, Latvia), including identified correlations and the need for an effective on-field training solution. A key focus is eliminating reliance on devices like mobile applications or screens, which can lead to incorrect training positions and other challenges. This chapter outlines the interdisciplinary approach to developing a DAid® smart socks-based biofeedback system concept by using an iterative designing approach.

*3.1. Methods*

Wearable technology and smart textiles in healthcare, as inherently multidisciplinary fields, integrate knowledge from disciplines such as textile design and architecture, biomedical engineering, biomechanics, healthcare, and data science. Accordingly, our research approach is interdisciplinary, intertwining methods from these collaborating fields. The roles and contributions of team members from various disciplines are outlined in Table 1.

This concept protocol focuses on the design process, utilizing iterative design methods [43]. The development of a proof-of-concept functional prototype for real-time biomechanical monitoring and feedback to enhance injury prevention in football consisted of

an iterative design process as a cyclical method (Figure 2) of developing and refining a DAid® smart socks-based biofeedback system through repeated cycles of brainstorming, prototyping, testing, and evaluation [43].

**Table 1.** Cross disciplinarity team roles.

| Cross-Disciplinary Area | Tasks During Conceptualization Process |
|---|---|
| Healthcare and Biomechanics | - Analyze and identify key injury prevention indicators for biomechanical in real-time monitoring.<br>- Participate in the development process of DAid® smart socks-based biofeedback system for injury prevention. |
| Sociotechnical Systems Modeling | - Model user interactions with the wearable system, focusing on usability in real-world settings.<br>- Assess the social and psychological factors that may affect the adoption and use of the system (e.g., motivation, trust).<br>- Ensure the system's design promotes long-term engagement and ease of use without causing distractions or discomfort during training. |
| Biomedical Engineering | - Discuss the hardware durable components (e.g., sensors) that will be embedded in the DAid® smart socks-based biofeedback system for real-time biomechanical monitoring.<br>- Ensure the sensors accurately capture key biomechanical data without interfering with the player's movement.<br>- Modeling of the prototype for accuracy, comfort, and reliability under different conditions (e.g., weather, movement types). |
| Data Science | - Propose algorithms to process the biomechanical data collected from the sensors.<br>- Develop models for real-time feedback based on biomechanical data (e.g., alerting players to foot posture corrections).<br>- Ensure the data collected can be securely stored and accessed for analysis. |

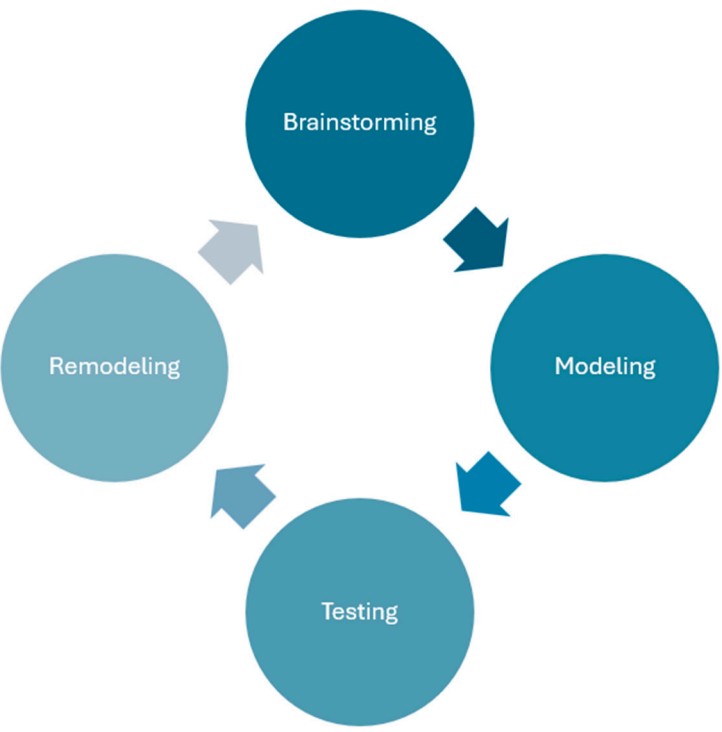

**Figure 2.** Iterative design process for DAid® smart socks-based biofeedback system development.

Key components of the iterative design process for DAid® smart socks-based biofeedback system development include the following:

- Brainstorming: Generating ideas and potential solutions for identified key injury prevention indicators for biomechanical data in real-time monitoring.
- Designing and modeling: Creating initial models for prototypes or drafts based on brainstormed ideas.
- Testing: Evaluating the concept models to identify strengths and weaknesses.
- Redesigning and remodeling: Making improvements and adjustments based on the testing feedback.
- Repeating: Continuing this cycle until the desired outcome is achieved.

As illustrated in Figure 2, the development process began with acquiring an understanding of the usage context and its demands through creative brainstorming sessions. These findings were then converted into the conceptual framework of the DAid® smart socks-based biofeedback system, with its features being evaluated by creating prototypes at various stages of refinement. During the initial ideation phase, qualitative information was gathered by engaging participants to share their experiences with conventional injury prevention programs like FIFA 11+, aiming to gain deeper insights into usability and wearability factors. In this context, usability pertains to the ease of operation of the DAid® smart socks-based biofeedback system in effectively accomplishing the intended objectives, while wearability evaluates the comfort and durability of the system under prolonged use on the football field during the prevention program execution.

The following subsections will describe the steps taken to create the DAid® smart socks-based biofeedback system concept starting from the brainstorming sessions.

### 3.2. Brainstorming Sessions

The development of the DAid® smart socks-based biofeedback system commenced with a comprehensive exploration of the needs and requirements for effective biomechanical monitoring in football training. To establish a solid foundation for the prototyping process, a series of brainstorming sessions were organized among the project team, which included researchers, physiotherapists, and technology developers. These sessions aimed to identify the key functionalities of the DAid® smart socks and address existing limitations in current wearable technologies related to data accuracy, user experience, and integration with mixed reality feedback systems. Initial discussions centered on the technical challenges of connecting multiple Bluetooth devices, leading to the decision to utilize a laptop for data processing rather than a tablet or smartphone.

During these sessions, various concepts for enhancing data acquisition were considered, including the potential to reduce the data acquisition rate and optimize sensor configurations within the socks. The team also addressed how feedback would be conveyed to participants, debating whether to utilize visual, audio, or a combination of both methods for real-time performance insights. Key considerations emerged regarding the durability of the socks under training conditions, particularly the impact of sweat on sensor performance and the need for rigorous testing to assess their resilience during high-intensity activities. The discussions culminated in the identification of specific requirements for the prototype, including the ability to monitor biomechanical parameters accurately, facilitate simultaneous use for an entire team, and provide tailored feedback based on individual exercise protocols. This collaborative effort set the stage for the next steps in the development process, aligning the team's vision and objectives for creating an effective injury prevention tool in football. The brainstorming sessions resulted in a list of the aspects (Figure 3) needed to be considered for a wholistic technological system, which are as follows:

- Technological requirements and limitations.
- User perception, football training, and motivational psychology.
- Human–computer interaction specific.
- Injury prevention training exercise content requirements and restrictions.
- Content transfer to the real-time feedback requirements and limitations.

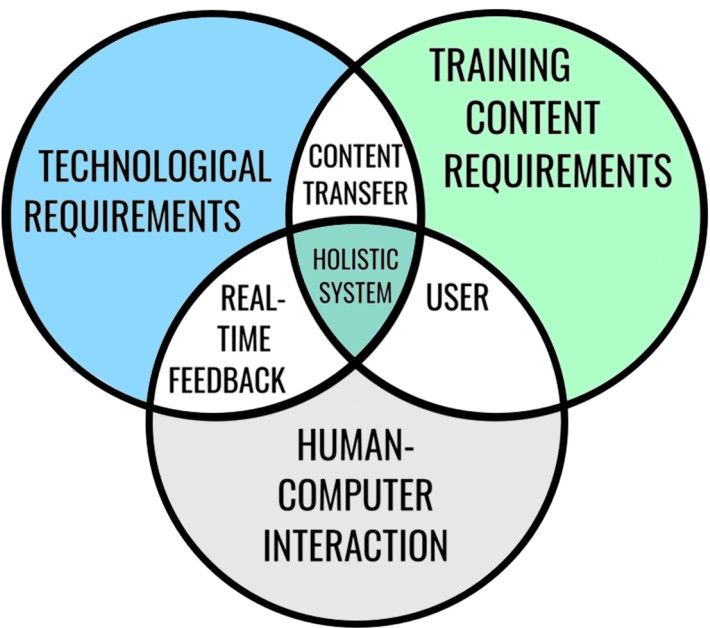

**Figure 3.** Aspects of DAid® smart socks-based biofeedback system development.

### 3.3. DAid® Smart Socks-Based Biofeedback System Parameter Selection

The DAid® smart socks-based biofeedback system requires precise parameter selection to facilitate accurate and meaningful insights during athletic performance. This system processes sensor data from smart socks, delivering real-time biofeedback through a head-mounted display (HMD) to provide athletes with immediate information about their movement patterns. Key metrics collected by smart socks during exercise include the coordinates of the center of pressure (COP1x, COP1y, COP2x, COP2y) which indicate foot pressure distribution and have demonstrated significant correlations with hip adduction and internal rotation, as identified in the study results [24]. Given the strong positive correlation between hip adduction and changes in COP1x (r = 0.785, *p* < 0.001), COP1x has been identified as highly responsive to deviations in movement, particularly knee valgus positioning during a single-leg squat [24]. This parameter exhibits high sensitivity to both correct and incorrect movement patterns, positioning it as a primary metric for real-time biofeedback. The DAid® smart socks system measures several key biomechanical indicators, including plantar pressure distribution, weight shifts, and the center of pressure (COP). COP changes provide insight into postural stability and lower limb mechanics, which are crucial for preventing excessive joint loading and injury. To ensure the real-time feedback in the headset based on COP changes, a custom system is developed using on .NET Framework 4.8 (further in the document mentioned as "system"). During system development and collaborative discussions, it was determined that visualizing COP1x changes would offer immediate, intuitive feedback on alignment and stability. Such visualization can effectively highlight shifts in medial–lateral foot pressure, guiding athletes in modifying their movements to optimize form. For example, if a player demonstrates excessive medial–lateral movement in their foot (a sign of poor ankle stability), the system signals an adjustment, allowing for more stable foot positioning and reduced stress on the knee and ankle. Focusing on COP1x as the central parameter for biofeedback visualization

enhances the system's capacity to support proper lower limb alignment, potentially reducing injury risk and enhancing performance, especially in the high-demand movements characteristic of football and other dynamic sports. By providing real-time feedback, the system enables athletes to identify and correct improper biomechanics immediately. This targeted parameter selection underscores the DAid® smart sock system's value in the real-time assessment and correction of biomechanics in athletic populations, empowering athletes to maintain optimal movement patterns and reduce the risk of injury over time.

### 3.3.1. Headset Choice and Considerations

The system integrates with a Meta Quest 3 headset (Meta Platforms, Inc., Menlo Park, CA, USA), chosen for its high throughput and low latency performance, important for delivering real-time biofeedback in virtual training sessions. Additionally, Quest 3's capability ensures stable data transfer, essential for maintaining the integrity of real-time sensor data. The decision to use the Meta Quest 3 also considers its improved screens and lenses with 2 LCD displays and improved resolution compared to previous models of Meta headsets, including access to a wide array of development tools and the ability to handle complex rendering tasks efficiently. This headset meets the primary need for a low-latency, high-quality MR experience, though it does rely heavily on Wi-Fi stability, which may present issues in environments with congested networks. A crucial aspect is its low weight in comparison to Quest 2 and other models. Even so the system will be usable on Quest 2 or Quest 3S or newer headsets which can meet all the necessary requirements.

### 3.3.2. Hardware Considerations

The brainstorming phase helped to figure out that there are multiple available options for the application part of the biofeedback system, namely a tablet, smartphone, and computer. The decision to develop the biofeedback system on a computer was based on several key factors that were figured out during the brainstorming sessions. While tablets and smartphones were considered options due to their portability and outdoor usability, the computer was chosen for its high processing power and ability to manage more complex tasks. First, while tablets offer larger screens and smartphones provide portability, both device types have limitations in terms of processing heavy data streams and running multiple tasks simultaneously. The computer, on the other hand, provides a higher level of processing power and can handle real-time data better, which is critical when working with continuous sensor input from the DAid® smart socks. Additionally, the computer offers a more robust environment for development, including support for advanced integrations, such as connecting with external hardware (e.g., HMD) or additional peripherals. This makes it easier to scale the system and add new features over time.

### 3.3.3. Information Flow

Overall information flow: For the intended application, in the first iteration there were up to three participants simultaneously using the system. Each participant wore a pair of smart socks, thus there were data flows from six socks simultaneously.

The number of issues were considered while researching different connection types between sensors and application, application, and headset. Different communication technologies exhibit varying levels of latency which can impact real-time applications. For instance, TCP (Transmission Control Protocol) can experience high latency due to delays in data transfer, which is critical in applications requiring immediate data updates, such as the given food plantar changes monitoring.

BLE (Bluetooth Low Energy) is optimized for low-latency communication, making it suitable for timely sensor data updates. However, in congested environments or with multiple devices connected, latency issues can still arise.

After performing the research, we decided to use Bluetooth. It was chosen for its higher bandwidth compared to Bluetooth Low Energy (BLE), making it more suitable for managing the larger volumes of sensor data generated during high-frequency monitoring of foot pressure and movement. The system requires the transmission of continuous, real-time data, with each sensor sending 3–4 data points per package, and Bluetooth can support this without the bandwidth limitations of BLE.

Wi-Fi was chosen as the main communication between the HMD and computer. However, Wi-Fi cannot be used as the data transfer source between sensors and computers as it can affect latency, impacting the speed at which data are displayed and commands are sent as high latency can disrupt applications needing real-time data.

This balance between Bluetooth for sensor communication and Wi-Fi for data transfer to the HMD ensures the system meets the real-time requirements of biofeedback applications while minimizing latent issues.

### 3.3.4. Operating System Consideration

Operating system consideration included three systems: Android, Windows, and Linux. It was decided to run the application on Windows, which supports the necessary Bluetooth and Wi-Fi capabilities for the DAid® smart socks-based biofeedback system. Windows was chosen for its broad compatibility with a variety of tablets and smartphones, ease of customization, and the availability of development tools that facilitate the integration of Bluetooth and MR functionalities. It also offers the flexibility to manage background services, which are critical for maintaining continuous Bluetooth connections with the sensors and ensuring real-time updates are sent to the HMD without disruption. The operating system must also ensure smooth handling of multi-tasking between data collection, processing, and MR integration, making it well suited for the demands of this system.

### 3.3.5. Technical Innovations and System Architecture for Multi-User Biomechanical Feedback

The DAid® smart socks-based biofeedback system tackles technical challenges through innovative design and engineering solutions.

The study concept is to manage the training process for three soccer players at the same time, wearing the HMDs which means managing the training process of all the players from one PC. The system must handle high-frequency sensor data transmission from six smart socks simultaneously, ensuring that data are processed, analyzed, and displayed on a head-mounted display (HMD) within a latency of less than 100 ms. This is tackled both by data processing logic and a system architecture approach. This requires robust data processing and hardware capable of supporting such high throughput. The use of a layered system architecture enables modular and efficient handling of tasks. Therefore, in order to avoid the processing capacity issues, which would arise if the data processing and data transfer were performed in the HMD, all the data processing and data transfer logic were conducted on the PC (client and server sides of the program) and only a minimum of commands and data were sent to the HMD. The data processing module employs a real-time parser to normalize raw data, compute center of pressure (COP) values, and pass results into the MR application. Additionally, the adoption of Bluetooth for sensor data and Wi-Fi/TCP for HMD communication strikes a balance between bandwidth and latency.

Simultaneously connecting and managing data from multiple Bluetooth channels for six socks while ensuring stable and interference-free connections in environments prone to congestion is complex. The system implements a dedicated BluetoothService to maintain robust connections, using reconnection logic to address potential disconnections. By optimizing the communication protocol and assigning specific Bluetooth addresses to each sensor, the system ensures seamless, concurrent data streams. Simultaneously

managing multiple Bluetooth connections for six smart socks in environments prone to interference and congestion requires innovative solutions. The system implements a multi-threaded communication management framework that assigns a dedicated thread to each Bluetooth connection. This framework incorporates a priority-based scheduling algorithm to prevent collisions, packet loss, or delays in data transmission. Additionally, an adaptive reconnection logic ensures that disconnections are handled seamlessly without impacting overall performance. For example, during exercises such as "vertical jumps", where the user has to jump as quickly as possible in different directions thus complicating the real-time feedback, the feedback uses adaptive reconnection logic, creating feedback out of data received in between the repetitions of the exercise. By leveraging these innovations, the system ensures reliable and interference-free communication across all six socks.

Creating an immersive MR environment on the Meta Quest 3 HMD (Meta Platforms, Inc., Menlo Park, CA, USA) that visualizes real-time feedback while maintaining high-quality rendering and low latency is technically demanding. The HMD must seamlessly integrate with the application to provide an engaging user experience. The Meta Quest 3 headset was selected for its rendering capabilities, low latency, and high-resolution displays as well as pass-through to reduce motion sickness and controller-free functionality for freedom of movement which is crucial for soccer players. The MRCommunicationService transforms sensor data into a format compatible with the HMD, enabling dynamic visualizations. This innovation allows real-time adjustments to be displayed in an interactive and visually engaging manner. The HMD application was developed to provide minimum processing but receive most of the needed calculations and feedback from the PC side. Therefore, it only receives commands, data, and shows the visualizations.

The architecture also allows for future scalability, enabling additional sensors or participants to be integrated with minimal reconfiguration. For example, adding accelerometer data to measure the angle between the foot and lower limb to ensure the back leg is bent in the knee joint during the lunge.

Coordinating data from multiple participants in real-time with little to no latency issues or data misalignment is critical for ensuring accurate and actionable feedback, especially due to the dynamic character of the specific FIFA11+ exercise program. The system's client–server architecture ensures synchronized data handling. The server receives sensor data, processes them, and transmits synchronized results to the client application. The PC application sends the data to the HMD with the live feedback in real time according to the training program, thus reducing the risk of receiving faulty data that may occur during pauses in between the exercises. This setup minimizes delays and ensures consistent feedback across all users.

Accurate calibration of the DAid® smart socks is essential to ensure reliable COP measurements and feedback. One of the challenges throughout the project is variability in sensor performance or user-specific biomechanics that can impact data accuracy. The system includes a calibration feature that adjusts sensor readings based on user-specific parameters. Calibration data are stored locally, allowing the system to compensate for individual differences and maintain accuracy through session. The calibration is adjusted due to the specific needs of the exercises, as some of the exercises require both legs at the same time and some only one at a time; therefore, the program needs to have two different calibrations in order to be further used as the default depending on the exercise.

The use of Bluetooth for sensor data transmission and Wi-Fi/TCP for HMD communication is optimized through a dual-layer communication architecture. The Bluetooth layer handles high-frequency data streams from the socks, while the Wi-Fi/TCP layer prioritizes processed data for real-time visualization on the HMD. The system incorporates adaptive data rate protocols for Wi-Fi communication, ensuring that bandwidth is utilized efficiently

and that latency remains under 100 ms. This hybrid communication strategy balances bandwidth and latency demands, ensuring seamless integration between the smart socks, server, and the HMD.

The team has held multiple brainstorming sessions on biomechanics, recognizing its critical impact on the system. These discussions focused on ensuring accurate data capture, personalized calibration, and precise center of pressure (COP) analysis, shaping the system's design and real-time feedback. The user interface must provide real-time feedback, be easy to navigate, and support complex data visualizations without overwhelming the user. The GUI and MR application were designed with user experience in mind. The MR interface delivers engaging visual feedback. Storing and retrieving high-frequency sensor data for future processing from the PC helps to minimize the processing that happens specifically on the HMD. High-resolution displays are essential for delivering clear and detailed feedback, ensuring users can accurately interpret biomechanical data such as center of pressure (COP) trajectories. The Meta Quest 3 HMD was chosen for its high-resolution displays and advanced rendering capabilities, which support dynamic visualizations without sacrificing clarity or precision.

### 3.3.6. Biomechanics-Driven Data Processing and Feedback Logic

The system's focus on biomechanics had an impact on the logic of the entire application, particularly in the way data were processed and feedback was generated. Real-time analysis of biomechanical parameters such as the center of pressure (COP) demanded that the system handle raw data streams with extreme precision and efficiency. The algorithms used to parse and normalize data were specifically designed to capture couple seconds variations in plantar pressure and adjust for individual user differences. This required implementing calibration logic within the system to make sensor outputs less "noisy" based on each user's unique biomechanics. It also influenced the logic for movement analysis. The system's code incorporates thresholds exclusive for each exercise to identify incorrect movements. This was achieved by testing in real life and finding out the optimal positions of the feet during given exercises, enabling the system to detect and flag deviations from initial movement patterns. For example, big shifts in COP values triggered alerts within the MR interface, signaling the user to correct their stance in real time. In order to align with the specific evidence-based FIFA11+ program, each exercise is assigned its own COP buffer zone which consists of green and red zones. This way when the user performs an exercise program, it can calculate whether the foot is positioned correctly, needs to be moved (if it is in the red zone), or it is out of the buffer zone and the exercise has to be restarted.

The system's modular architecture played a key role in supporting these biomechanical requirements. Separate modules for data parsing, movement analysis, and MR visualization allowed developers to continuously refine each component without disrupting the overall functionality. The focus on biomechanics shaped not just the functionality of the system but also its underlying principle, ensuring that every technical decision was aligned with the goal of providing accurate, actionable feedback for enhanced movement performance and injury prevention during the exercises.

### 3.4. Planned Study Population for Testing DAid® Smart Socks-Based Biofeedback System

The study included eighty youth soccer players, divided evenly into two groups (an active comparator group and experimental group) of forty participants each. The sample size was calculated using G*Power version 3.1.9.7, aiming to detect an effect size greater than 0.5, with an alpha level of 0.05 and a study power of 0.95. All participants were recruited through the Latvian Football Federation, and confidentiality was ensured by assigning each participant a unique identification number instead of using names.

Inclusion and Exclusion Criteria

Inclusion Criteria:

- Players actively participating in the Youth Soccer League.
- Age between 14 and 18 years.
- Parental consent is required for participants under eighteen.
- Players who have completed at least three months of the FIFA 11+ program before the study.
- Agreement to participate in the study and ability to understand and speak Latvian.

Exclusion Criteria:

- Knee or ankle pain rated above 5 on the VAS (Visual Analog Scale) for pain or an increase of more than two points during a single-leg squat.
- History of significant musculoskeletal injury within the past six months.
- Concurrent participation in other injury prevention programs or research trials.
- Medical conditions that contraindicate physical activity.
- Recent lower limb surgery (within the past nine months).
- Lower limb injuries within the last six months.
- Vestibular disorders, metallic implants in the sensor or head-mounted display (HMD) application areas, current eye infection, or photosensitivity (including risk of epilepsy).

*3.5. Study Procedures for Participants Who Will Use the DAid® Smart Socks-Based Biofeedback System*

The structured intervention will utilize the FIFA 11+ injury prevention program over a 12-week period, consisting of two 45 min sessions per week. The exercises will target improvements in strength, balance, proprioception, and dynamic stability in the lower limbs. The procedures are delineated as follows: the active comparator group will perform standard FIFA 11+ program exercises without the DAid® smart socks-based biofeedback system. In contrast, the experimental group will participate in selected FIFA 11+ exercises enhanced by real-time biofeedback from the DAid® smart socks system. Each session will begin with a warm-up of light jogging and dynamic movements. During the specific exercises outlined in the FIFA 11+ manual, First Level, Part 2 (Exercises 10, 11, and 12), Second Level, Part 2 (Exercises 11 and 12), and Third Level, Part 2 (Exercises 11 and 12) participants will wear the DAid® smart socks, which continuously monitor plantar pressure and the center of pressure (COP). This system transmits real-time data to a head-mounted display (HMD), allowing participants to make immediate adjustments to their foot movements to optimize form and alignment. For instance, if a participant's foot positioning is incorrect, a red arrow will appear on the HMD, indicating the need for correction, such as addressing an inward foot roll. Following the warm-up, participants will engage in biofeedback-integrated exercises, receiving real-time feedback on plantar pressure and COP metrics. This feedback will facilitate immediate adjustments to balance, weight distribution, and limb positioning, particularly during key exercises such as the single-leg stance, squats, and lateral jumps. This capability will enable participants to refine their techniques and correct alignment issues effectively. While the experimental group will utilize biofeedback for select exercises, they will also perform agility and coordination drills without feedback, mirroring the active comparator group. This design ensures consistency across both groups regarding the overall structure of the training program. Each session will similarly conclude with a cool-down involving light jogging and recovery exercises to promote gradual heart rate reduction and facilitate recovery. This structured approach, utilizing the DAid® smart socks system for selected exercises, ensures that the experimental group effectively leverages biofeedback to enhance performance while minimizing injury

risk, all while participating in other components of the program without feedback, thus maintaining alignment with the active comparator group.

*3.6. Data Analysis Methods and Statistical Considerations*

The data analysis methods and statistical considerations for the study will focus on evaluating the effectiveness of the DAid® smart socks-based biofeedback system in enhancing injury prevention and improving performance among youth soccer players. Specifically, we aim to compare the injury rates, biomechanical improvements, and overall performance between the experimental group (using the DAid® biofeedback system) and the active comparator group (using the standard FIFA 11+ program without biofeedback). The analysis will involve both quantitative and qualitative data, and the primary outcomes will include injury incidence, biomechanical parameters, and exercise performance metrics.

### 3.6.1. Primary Outcome Measures

The primary outcome, improvement in Functional Movement Screen (FMS) scores, will be analyzed using paired *t*-tests or Wilcoxon signed-rank tests, depending on the normality of the data, to assess changes from baseline to the 12-week intervention within each group. Between-group comparisons will be performed using independent *t*-tests or Mann–Whitney U tests for non-normally distributed data.

Real-time biofeedback from the DAid® smart socks system will provide continuous monitoring of foot pressure distribution and the center of pressure (COP). Key parameters such as COP1x (medial–lateral movement), COP1y (anterior–posterior movement), and other relevant metrics will be used to assess the participants' movement patterns, particularly during exercises like the single-leg stance, squats, and lateral jumps. These parameters will be compared between both groups to assess improvements in biomechanics, alignment, and movement quality.

### 3.6.2. Secondary Outcome Measures

Injury frequency data will be collected using a standardized injury reporting form. Injury events will be documented, including the type, location, and severity of each injury, with severity measured using the Visual Analog Scale (VAS) for pain. Injury frequency will be analyzed at the one-year follow-up by comparing the number of in-juries per 1000 h of play between the active comparator and experimental groups using Poisson regression models.

### 3.6.3. Planned Statistical Considerations for the Concept Validation

Descriptive statistics (mean, standard deviation, and range) will be used to summarize the baseline characteristics of the study participants, including age, gender, injury history, and pre-intervention performance measures. For within-group comparisons, changes from baseline to post-intervention will be assessed using paired *t*-tests or Wilcoxon signed-rank tests, depending on the normality of the data. For between-group comparisons, differences between the experimental group (DAid® biofeedback system) and the active comparator group (FIFA 11+ program) will be analyzed using independent *t*-tests (for normally distributed data) or Mann–Whitney U tests (for non-normally distributed data). These comparisons will assess differences in injury rates, biomechanical changes, and performance improvements. Repeated measures Analysis of Variance (ANOVA) will be used to assess changes over time (e.g., pre-intervention vs. post-intervention) for continuous outcomes like biomechanical parameters (COP1x, COP1y) and performance metrics, considering within-subject correlations and tracking changes across multiple time points. Cohen's d will be calculated to measure the magnitude of differences between groups, particularly for key performance and biomechanical outcomes. An effect size

greater than 0.8 will be considered large, indicating a significant practical difference between the groups. Pearson or Spearman correlation tests will be used to assess the relationships between biomechanical parameters (e.g., COP1x) and injury rates. A strong negative correlation would suggest that improvements in biomechanical alignment are associated with lower injury incidence. If necessary, multivariate regression models will be used to adjust for potential confounding variables, such as baseline performance levels, age, gender, or previous injury history. This will allow for a more comprehensive understanding of the relationship between the DAid® biofeedback system and injury prevention outcomes.

### 3.6.4. Data Quality and Monitoring

To ensure the reliability of the data, all DAid® smart socks will undergo calibration prior to use, with ongoing checks during each session to ensure accurate data collection. Assessors will be blinded to group assignments to minimize bias in performance and injury assessments. Data transfer from sensors to the system will be monitored in real-time to ensure no data loss. Any missing or incomplete data will be handled using imputation methods if necessary.

## 4. Concept of a DAid® Smart Socks-Based Biofeedback System

The DAid® smart socks-based biofeedback system is designed to provide real-time feedback on a user's biomechanical performance through an MR interface, using data transmitted from smart sensors embedded in the socks. The system's architecture relies on two primary communication technologies: Bluetooth for sensor data transmission and Wi-Fi for MR streaming.

### 4.1. System Components

The application uses layered architecture that includes the following components:

- Presentation Layer: Handles the UI and user interactions.
- Domain Layer: Manages logic and data processing.
- Data Layer: Responsible for data acquisition from sensors, storage, and network operations.
- Communication Layer: Manages Bluetooth communication with smart socks and data transmission to the HMD.

Components of the architecture can be seen in the scheme in Figure 4.

### 4.2. System Activities and Fragments

Activities of the system consist of Main Activity and Statistics Fragment. Main Activity acts as the main entry point that displays the dashboard with current training sessions and connection status to smart socks. It initializes Bluetooth service and connects to sensors on creation, while additionally checking the connection status and refreshing UI. Statistics Fragment displays real time analyses of players' movements; it receives live data from the sensors and handles the UI updates with graphs and tables. The system's architecture has been developed according to the chosen connection types, Figure 5.

1. DAid® smart socks—Wireless sensors that collect plantar pressure data and transmit it over Bluetooth to the server, which is represented by the computer.
2. Server—Computer acts as a server that receives the sensor data over Bluetooth. It performs initial data receiving by opening Bluetooth serial connection, sending the command to start sending data for the sensors. In each iteration, the server checks if any data have been received (serialPort. BytesToRead > 0). If data are available, it reads the data, processes each packet, validates it, and stores the normalized results. Since the server and the client (step 3) are both on the same computer, the data remains on the same machine, simplifying communication.

3. Client—Client periodically requests the latest data from the server on the same computer. The client acts as an intermediary for the Graphical User Interface, enabling it to visualize the data like pressure readings, center of pressure and calibration status, or any warnings that the server may have found. The client raises an event every time new data are received, which can be used for real-time feedback and updates in the GUI application (step 4) and MR application.

4. Application—The Graphical User Interface (GUI) on the same computer displays an application, which is the initial part of the biofeedback solution. The GUI initializes and establishes a connection with the client–server system. It offers controls that allow users to start or stop data collection from the socks. This interface allows users to insert their data and choose the complexity level of their workout as well as starting it. It sends the data needed for the feedback in the HMD. The input data of the application are sent back to the client side, which proceeds with sending them to the HMD.

5. MR application—Receives data from the computer's application. It uses the processed data to create an immersive environment where the feedback is being visualized, providing engaging and interactive experiences, such as VR-based training simulations. It gives details on the changed foot plantar pressure, helping to correct it.

6. Data transfer over Bluetooth—The DAid® smart socks device is connected to the server via the COM port by Bluetooth. The server begins by sending a command over the serial connection "BTˆSTART". This command instructs the smart socks to start streaming data packets. Each data packet is sent 10–15 times per second, containing around 3–4 data points. The server continuously listens for incoming data on the COM port and reads them into the buffer. Then, it scans the buffer to detect packets that start with the Start Byte and end with the Stop Byte. Each sensor's value can be represented by two bytes, so for six sensors, the data section would be around 12 bytes.

7. Data Transfer over WiFi—The client side establishes a connection over TCP/IP with the head-mounted device and by the command received from the GUI application side (e.g., "Start"), starts to send sensor data such as the COP to the HMD.

*4.3. System Requirements and Data*

The system relies on several core services and modules to ensure seamless data transmission, processing, and storage. The BluetoothService manages all Bluetooth communication with the DAid® smart socks. Its primary function is to scan for available devices, establish connections, and continuously read data from the smart socks. The service handles potential disconnections by implementing reconnection logic to ensure the system remains operational during training sessions. Key methods include connecting to the smart socks via a specific Bluetooth address and continuously retrieving sensor data in real-time. The MRCommunicationService is responsible for sending processed data to the HMD over a Wi-Fi network. Once the sensor data are received and processed by the system, they are converted into a MR-compatible format before being transmitted to the HMD. This service ensures the reliability of data transmission to maintain the accuracy and consistency of real-time feedback during the training sessions. The DataProcessingModule is tasked with analyzing raw data collected from the smart socks. It involves three key steps: data parsing, movement analysis, and data aggregation. The system includes a local storage solution for storing both raw and processed data. The storage system is structured in a database with two primary tables: Sessions, which holds metadata about each training session (e.g., start time, duration), and SensorData, which contains both raw and processed sensor data linked to specific sessions. This allows for later retrieval and analysis, providing users with

the ability to review past performance and monitor progress over time. These services and modules work together to ensure that sensor data are accurately captured, processed, and transmitted to the MR environment.

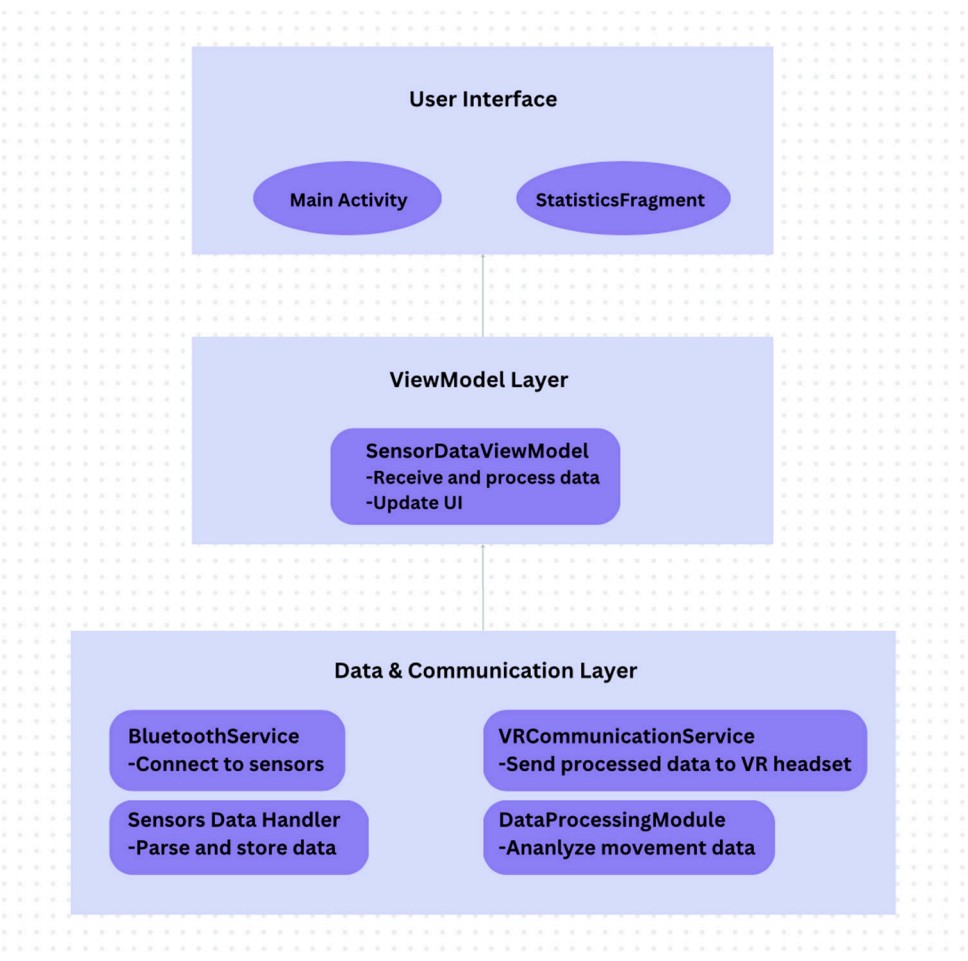

**Figure 4.** Component diagram for the biofeedback system application.

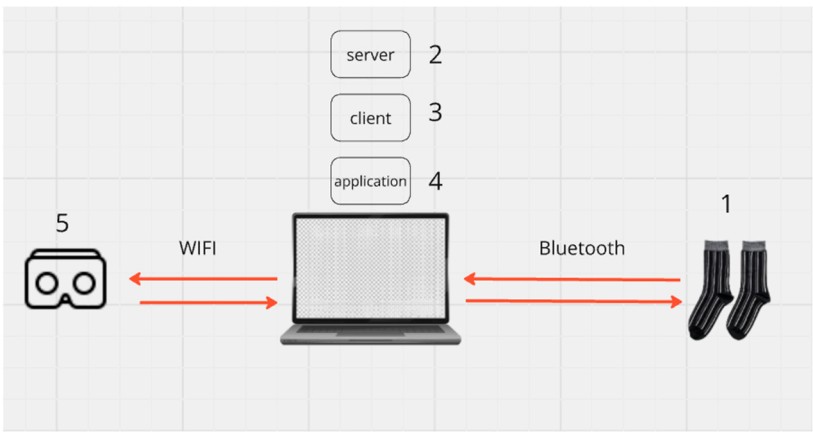

**Figure 5.** System architecture of the biofeedback system.

### 4.4. System Requirements for the Concept of DAid® Smart Socks-Based Biofeedback System

The smart socks are equipped with pressure sensors on both legs, each transmitting data via Bluetooth over two separate channels to a computer or other processing device. Each sensor package contains 3–4 data points, including metrics such as plantar pressure

distribution and other key movement indicators. Sensor data are transferred over Bluetooth to the server.

Once the sensor data are received by the application, they are processed in real time and then transmitted to the Meta Quest 3 HMD over a Wi-Fi connection. This allows the user to receive immersive feedback through MR, enabling them to adjust their movements based on the system's real-time analysis. The combination of Bluetooth for sensor data and Wi-Fi for MR streaming provides a balance between the need for low-latency, high-speed data transmission and power efficiency.

*4.5. Requirements for the First Prototype—Minimal Viable Product (MVP) of DAid® Smart Socks-Based Biofeedback System*

A Minimal Viable Product has been created that consists of three components: server, client, and DAid® smart socks. The MPV architecture is presented in Figure 6.

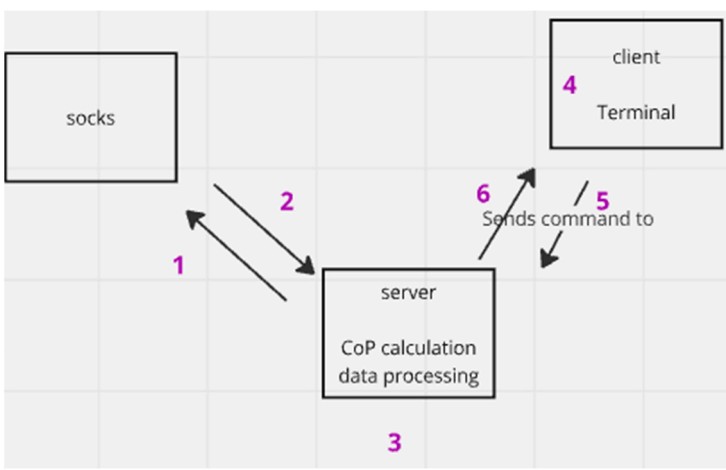

**Figure 6.** MVP system architecture of the biofeedback system.

There are six steps in the MVP system architecture:

1. Command to Start Data Flow: The system sends an initial command to the sensor socks to activate data transmission, initiating the flow of sensor data.
2. Sending Data from the DAid@ Smart Socks: Data received continuously from the DAid@ smart socks, maintaining an ongoing communication loop.
3. Data Reception and Processing: The system receives raw data packets from the socks, converts them into numerical values, normalizes the data, writes sensor-specific information to a CSV file, and computes real-time center of pressure (COP) coordinates (x, y).
4. User Input Commands: The user interacts with the system commands via terminal sending such as "connect", "calibrate", and "start" enabling connection to the socks, calibration of the sensors, and initiation of data collection.
5. Command to Start Receiving Data: A specific command is issued to the server to begin receiving and processing sensor data from the socks.
6. Sending COP Data to Client: The processed COP data are transmitted to the client side, where they are displayed on the terminal in real time for monitoring and feedback.

The system must support a data sampling rate of 200 Hz, ensuring that the sensor data are collected frequently enough to provide a smooth and accurate reflection of foot pressure and movement changes. Data transmission from the sensors occurs 10–15 times per second, and a 1 MB buffer is recommended for managing the incoming data streams efficiently. This buffer uses a circular data handling approach, preventing data loss or overflow during periods of intense data transmission.

Regarding hardware requirements, the application requires a multi-core processor with at least four cores and 8 GB of RAM (with 16 GB being ideal) to handle the data analysis and multiple data streams. The system also needs to maintain a feedback latency of less than 100 ms to ensure that users receive actionable feedback promptly.

## 5. Discussion

### 5.1. Introduction to the DAid® Smart Socks for Injury Prevention

This study outlines the development and early implementation of a biofeedback system using DAid® smart socks for sports injury prevention. By exploring how wearable biofeedback systems can be incorporated into structured injury prevention programs, this research contributes to the advancement of technology-driven approaches to optimizing athletic performance and reducing injury risks [44].

The DAid® smart socks are being designed to provide real-time data on foot pressure, offering insights into movement patterns that could help prevent injuries in soccer players [22–24].

### 5.2. Addressing Gaps in Current Injury Prevention Programs

Existing injury prevention programs like FIFA 11+ have proven effective in reducing lower limb injuries in soccer by incorporating a structured warm-up routine and exercises targeting balance, strength, and flexibility. However, while FIFA 11+ helps improve general movement patterns, it lacks the ability to provide real-time, individualized biomechanical feedback, which is crucial for preventing injury due to improper foot mechanics or misalignments. Furthermore, FIFA 11+ relies on predetermined exercises, which may not fully address the dynamic and individualized needs of athletes in real-time training environments [11,45,46]. DAid® smart socks address this gap by continuously monitoring foot pressure and providing instant feedback, helping athletes make precise corrections during their training. Once fully developed and integrated with the FIFA 11+ program, the system is expected to enhance movement accuracy by tracking key metrics such as foot pressure and the center of pressure (COP) [11,24]. Real-time feedback has the potential to help athletes adjust their movements immediately, correcting misalignments such as improper foot rolls or uneven weight distribution. These capabilities are particularly relevant during exercises like squats, plyometric jumps, and single-leg stance, where improper technique often leads to overuse injuries, muscle imbalances, and joint stress [44,45].

### 5.3. Biomechanical Feedback for Injury Risk Reduction

Improved biomechanics directly reduce injury rates by addressing movement patterns that place excessive strain on the body. Research has shown that lower limb injuries, such as ACL tears and ankle sprains, are often caused by improper biomechanics, including issues like valgus knee collapse, excessive foot pronation, or uneven weight distribution during dynamic movements [46]. Unlike traditional injury prevention programs that offer static guidance, the DAid® smart socks system provides continuous biomechanical data, enabling real-time monitoring and feedback. This ongoing data stream tracks key metrics such as foot pressure and the center of pressure, allowing athletes to immediately adjust their movements—whether it is adjusting foot alignment or redistributing weight—reducing the risk of injury by ensuring optimal biomechanics. Research has shown that biofeedback-based interventions significantly improve movement efficiency and reduce non-contact injuries [46,47]. By offering continuous biomechanical monitoring and real-time corrections, the DAid® system ensures athletes are able to recognize and correct high-risk movement patterns as they occur, which potentially minimizes the likelihood of injuries. This approach is supported by existing studies, which have demonstrated that biofeedback-based inter-

ventions can significantly improve lower limb alignment and postural stability, leading to a decrease in non-contact injuries [47]. The integration of this technology with established injury prevention programs, such as FIFA 11+, can further enhance training effectiveness by ensuring athletes are moving in ways that optimize performance while minimizing injury risk.

*5.4. Technological Innovation: Wearable Tech and Mixed Reality*

This research is highly innovative because it integrates wearable technology, real-time biomechanical feedback, and mixed reality, creating a comprehensive system for injury prevention and performance enhancement. Unlike static, generalized approaches like FIFA 11+, DAid® offers dynamic, personalized feedback in real-time. The potential impact of this system is far-reaching—it could revolutionize how athletes across sports like soccer, basketball, and tennis train, rehabilitate, and optimize performance. By providing individualized biomechanical feedback, the system could reduce injury risks and improve movement efficiency, agility, and coordination, ultimately enhancing long-term athletic performance and recovery.

As part of its development, the system incorporates mixed reality (MR) to enhance the biofeedback experience [48]. The Meta Quest 3 headset is intended to deliver visual cues in an immersive, hands-free format, allowing athletes to focus on their movements rather than external devices. The system is being designed to support multiple participants simultaneously, making it suitable for team sports like soccer. If successfully implemented, this approach could provide real-time insights into both individual and team movement patterns, helping to optimize performance and reduce injury risks. The planned integration with the FIFA 11+ program aims to further improve biomechanics by enhancing balance, coordination, agility, and power. Continuous monitoring of foot and lower limb alignment may also contribute to the prevention of chronic injuries such as tendonitis and ACL-related issues [44].

*5.5. Future Directions and Challenges*

A key objective of this research is to develop a biofeedback system that supports a proactive approach to injury prevention. By identifying and correcting biomechanical deficiencies before they lead to injuries, the system could align with a preventive care model that shifts the focus from reactive treatment to early intervention. If effective, this strategy has the potential to reduce healthcare costs associated with sports injuries while improving overall athletic performance. Providing athletes and coaches with real-time data on movement inefficiencies may support modern training methodologies that prioritize injury prevention and long-term physical well-being.

However, several technical challenges must be addressed during development to ensure the system's reliability and usability. One potential limitation is the reliance on Bluetooth communication, which may be susceptible to interference or bandwidth constraints in environments with high device density. Future iterations of the DAid® system may need to explore alternative communication technologies, such as ultra-wideband (UWB) or advancements in Bluetooth protocols, to improve data transmission robustness.

Expanding the application of this biofeedback system beyond soccer to other sports, such as basketball, tennis, or track and field, could further increase its impact. Future research should explore the feasibility of monitoring additional biomechanical parameters, including joint angles, muscle activation levels, and fatigue metrics, to provide sport-specific feedback. Additionally, integrating personalized biofeedback tailored to an athlete's unique biomechanics, injury history, and learning pace could enhance user engagement and compliance, making interventions more effective at an individual level.

Another important area for future exploration is the potential for remote monitoring through cloud-based platforms. This capability could allow healthcare providers and coaches to oversee training sessions in real time or analyze recorded data for longitudinal tracking. By making advanced biofeedback tools accessible to remote or underserved areas, this approach could expand the reach of injury prevention technologies to a broader population of athletes.

This study establishes a concept for future evaluation of the feasibility and effectiveness of a smart socks-based biofeedback system in sports injury prevention and performance optimization. By combining wearable technology, real-time data analysis, and mixed reality feedback, this research contributes to the advancement of technology-driven injury prevention strategies. Addressing key challenges related to system connectivity, personalization, and sport-specific applications will be critical in refining the system and realizing its full potential in sports science and rehabilitation.

## 6. Conclusions

The DAid® smart socks-based biofeedback system offers a promising technological advancement in sports injury prevention and performance optimization. By integrating real-time biomechanical feedback into the FIFA 11+ injury prevention program, the system has the potential to reduce the risk of lower limb injuries in soccer players while enhancing movement quality and performance. Through the combination of wearable technology, mixed reality feedback, and precise biomechanical monitoring, this system represents an innovative approach to athletic training, with the potential for broader application in other sports and training settings. As technology matures and is further refined, it could play a significant role in revolutionizing injury prevention strategies in sport, offering athletes a more initiative-taking, data-driven approach to training and performance enhancement.

In the long term, biomechanics-focused feedback not only reduces injury risks but also improves movement quality, agility, balance, and force distribution, all key to peak performance. By continuously addressing biomechanical inefficiencies, athletes can optimize technique and support long-term physical health.

Future research should examine the system's long-term impact on injury prevention, explore personalized feedback based on individual biomechanics, and assess its effectiveness in youth athletes. Comparative studies with traditional programs, expansion to other sports, and cost-effectiveness analyses will help validate its potential. As the technology advances, it could significantly transform injury prevention and athletic performance training.

**Author Contributions:** Conceptualization, G.S., A.D., L.K., L.L., A.O., S.T., A.K. and L.L.; methodology, G.S., A.D., L.K. and L.L.; software, L.K. and L.L.; validation, G.S., A.D. and A.O.; formal analysis, G.S., A.D., A.O., A.K. and L.L.; investigation, L.K. and L.L.; resources, L.K. and L.L.; writing—original draft preparation, G.S., A.D., L.K., L.L. and A.J.; writing—review and editing, A.O., A.K., S.T. and S.D.; visualization, G.S. and L.K.; supervision, A.O., S.T. and A.K.; project administration, G.S.; funding acquisition, G.S., A.K., A.D., L.L., A.O. and S.T. All authors have read and agreed to the published version of the manuscript.

**Funding:** This research is funded by the Latvian Council of Science, project Smart textile solutions as biofeedback method for injury prevention for Latvian football youth league players, project No. lzp-2023/1-0027.

**Institutional Review Board Statement:** The study will be conducted in compliance with the Declaration of Helsinki and Latvian laws and adhered to the European Union's General Data Protection Regulation (GDPR) 2016/679. Prior to participation, the study's procedures will be thoroughly explained to participants and their guardians, and informed consent will be obtained. Ethical approval for the study was gained on 21 March 2023 by the Riga Stradins University Research Ethics Committee

(Approval No. 2-PĒK-4/294/2023). Participation in the study will be voluntary, with participants having the right to withdraw at any time without consequence. Participant confidentiality will be maintained throughout the study by assigning each participant a unique identification number, and all data will be anonymized for analysis and publication.

**Informed Consent Statement:** Informed consent was obtained from all subjects involved in the study.

**Data Availability Statement:** The data related to the project Smart textile solutions as biofeedback method for injury prevention for Latvian football youth league players, project No. lzp-2023/1-0027 are available from the corresponding author upon request. Data management plan identification number: DOI: 10.5281/zenodo.11082395 (accessed on 4 February 2024).

**Acknowledgments:** This research is inspired by the dedication and resilience of youth football players and young athletes. Their experiences and insights have been instrumental in shaping the focus of this concept protocol, emphasizing the need for innovative approaches in injury prevention and performance monitoring.

**Conflicts of Interest:** The authors declare no conflicts of interest. The funders had no role in the design of the study; in the collection, analyses, or interpretation of data; in the writing of the manuscript; or in the decision to publish the results.

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
