# Peer review of "Concept Protocol for Developing a DAid® Smart Socks-Based Biofeedback System: Enhancing Injury Prevention in Football Through Real-Time Biomechanical Monitoring and Mixed Reality Feedback"

_applsci, doi:10.3390/app15031584_

Round 1
Reviewer 1 Report
Comments and Suggestions for Authors
I believe this paper demonstrates considerable potential and innovation, yet there are still areas that can be improved and optimized. Below are my overall suggestions for revision:
1)Provide a detailed description of the advantages and disadvantages of existing prevention strategies (such as the FIFA 11+ program) to highlight the necessity of this research. Incorporate content from literature reviews to support the limitations of existing strategies.
2)Emphasize that the system is capable of providing continuous biomechanical data and facilitating real-time feedback and adjustments. Mention the potential impact of the system on improving biomechanics and reducing injury rates.
3)Stress the innovativeness of the research and its broad implications for sports injury prevention, rehabilitation, and athletic performance training. Include some outlooks on potential application scenarios.
4)Avoid using excessive specialized terminology or abbreviations unless they are crucial for understanding the paper. Simplify sentence structures to make the expression more fluent and natural.
Author Response
Comments 1: Provide a detailed description of the advantages and disadvantages of existing prevention strategies (such as the FIFA 11+ program) to highlight the necessity of this research. Incorporate content from literature reviews to support the limitations of existing strategies.
Response 1: Thank you for your insightful suggestion. We have incorporated a detailed discussion of the existing prevention strategies, including the FIFA 11+ program, in the revised manuscript. Specifically, we now provide a thorough literature review that outlines the advantages and limitations of these strategies, highlighting the gaps in current methodologies and demonstrating the necessity for the research presented in this paper. The revision can be found on page 2, paragraph 6, lines 53–60, where we describe the FIFA 11+ program’s structured approach to injury prevention, its benefits in improving neuromuscular control and balance, and its limitations in providing real-time biomechanical feedback.
Comments 2: Emphasize that the system is capable of providing continuous biomechanical data and facilitating real-time feedback and adjustments. Mention the potential impact of the system on improving biomechanics and reducing injury rates.
Response 2: We appreciate your input on emphasizing the system's continuous biomechanical data collection and real-time feedback capabilities. In the revised manuscript, we have dedicated page 10, paragraph 2, lines 366–378 to explain how our system facilitates ongoing biomechanical monitoring and provides immediate feedback for adjustments. This feature ensures that athletes can correct movement errors in real time, improving biomechanics and reducing injury risk. The revised text highlights how the DAid® smart socks track plantar pressure, weight distribution, and center of pressure (COP), transmitting this data to a mixed-reality headset for instant feedback, allowing users to adjust their movements accordingly.
Comments 3: Stress the innovativeness of the research and its broad implications for sports injury prevention, rehabilitation, and athletic performance training. Include some outlooks on potential application scenarios.
Response 3: Thank you for your suggestion to further emphasize the innovation of our research. We have revised the manuscript to stress the unique aspects of our approach, particularly its potential for improving sports injury prevention, rehabilitation, and performance enhancement. This revision is in page 20, paragraph 3, lines 796–811, where we elaborate on how our biofeedback system integrates smart textiles, real-time monitoring, and mixed reality to create a novel solution for injury prevention. Additionally, we have included potential application scenarios, such as implementation in professional sports teams, athletic training centers, and youth development programs, to illustrate the broader impact of our research.
Comments 4: Avoid using excessive specialized terminology or abbreviations unless they are crucial for understanding the paper. Simplify sentence structures to make the expression more fluent and natural.
Response 4: We appreciate your advice on simplifying the language. In response, we have revised the manuscript to reduce the use of specialized terminology and to make the sentence structures more fluent and accessible. The most notable changes are in page 12, paragraph 2, lines 453–468, where we simplified the explanation of how the biofeedback system processes and provides real-time movement corrections. Complex phrases have been reworded, and unnecessary jargon has been minimized to ensure clarity for a broader audience. These revisions enhance readability while maintaining scientific rigor.
Reviewer 2 Report
Comments and Suggestions for Authors
The manuscript is a protocol (or research proposal) for the development of a biofeedback system based on smart socks. The idea is interesting and the protocol is well written however, the following comments are provided:
1. Several portions of the text are written as if the system exist (or has been developed) and has been tested. Particularly the Abstract needs to be rewritten in order to better reflect the content of the manuscript, wich is a protocol and not a research paper.
2. The protocol lacks information about the hardware challenges (or necessary innovations) of the proposed design. Also, there is no discussion about biomechanics. In the opinion of this reviewer, these two topics should be better discussed and expanded in the protocol, as are the main areas of work and research that would actually enable the creation of the proposed device. If this were a research proposal submitted for funding, it would lack the necessary information to show that the authors have the technical expertise (both in terms of circuits, systems, and biomechanics) to succesfully complete the project.
Author Response
Comments 1: Several portions of the text are written as if the system exist (or has been developed) and has been tested. Particularly the Abstract needs to be rewritten in order to better reflect the content of the manuscript, wich is a protocol and not a research paper.
Response 1: We sincerely appreciate your valuable feedback. We acknowledge that certain portions of the manuscript were written in a way that might suggest the system has already been developed and tested. To address this, we have carefully revised the Abstract and other relevant sections to explicitly clarify that this is a protocol describing the design and development plan for a biofeedback system based on smart socks. Specifically, we have reworded the Abstract to ensure that it accurately reflects the nature of the manuscript as a research protocol rather than a completed study. The revision can be found in page 1, paragraph 2, lines 15–27, where we now explicitly emphasize that the study outlines the conceptual framework and planned methodologies for the development and evaluation of the DAid® smart socks system, rather than presenting tested results.
Comments 2: The protocol lacks information about the hardware challenges (or necessary innovations) of the proposed design. Also, there is no discussion about biomechanics. In the opinion of this reviewer, these two topics should be better discussed and expanded in the protocol, as are the main areas of work and research that would actually enable the creation of the proposed device. If this were a research proposal submitted for funding, it would lack the necessary information to show that the authors have the technical expertise (both in terms of circuits, systems, and biomechanics) to succesfully complete the project.
Response 2: Thank you for your valuable feedback. We have taken your comments into account and made significant revisions to address both the hardware challenges and biomechanics involved in the development of the DAid® smart socks system by adding Sections 3.3.5 and 3.3.6. In Section 3.3.5 (page 10, paragraph 3, lines 366–378), we discuss the technical innovations necessary for managing real-time, high-frequency sensor data from six smart socks while maintaining latency below 100 ms. We have included a more detailed discussion on potential issues related to Bluetooth communication, particularly its susceptibility to interference and bandwidth constraints in environments with multiple devices. To ensure reliable data transmission, we detail the implementation of a multi-threaded communication framework that assigns dedicated threads to each sensor and employs adaptive reconnection logic. We also discuss potential solutions, such as exploring alternative communication technologies like ultra-wideband (UWB) or improvements in Bluetooth protocols to enhance system performance in real-world conditions. Additionally, we justify the selection of the Meta Quest 3 HMD for its low latency, high-resolution display, and controller-free operation, which are crucial for real-time feedback in soccer training. In Section 3.3.6 (page 11, paragraph 2, lines 453–468), we significantly expand the discussion on biomechanics, emphasizing its critical role in the effectiveness of the DAid® system. We elaborate on how the system monitors key biomechanical parameters, such as foot pressure and the Center of Pressure (COP), to provide real-time feedback that helps athletes refine movement patterns and reduce the risk of injury. We explain how the system identifies and addresses improper movement patterns, such as excessive foot pronation, misaligned knee movements, and uneven weight distribution, which are commonly linked to lower limb injuries like ACL tears and ankle sprains. Furthermore, we describe the custom calibration logic integrated into the system to ensure accurate COP analysis across different exercises. These revisions strengthen our protocol by providing a more detailed discussion of the technical innovations and biomechanics-driven feedback logic, demonstrating our expertise in both areas. We appreciate your insightful comments and believe these additions provide the necessary depth to support the feasibility and impact of our research.
Reviewer 3 Report
Comments and Suggestions for Authors
This paper aims to enhance injury prevention in football through real-time biomechanical monitoring and mixed reality feedback. Personally, I think the topic is very interesting and meaningful. Before recommending the paper for publication, some suggestions are given as follows:
1. It would be better for the authors to give more detailed explanations about the used terminologies, such as FIFA 11+ program. These terms may be not familiar for readers.
2. The organization is good. However, I prefer a shorter background and related work. Currently, Section 2 is too tedious.
3. The authors are suggested to give a graphical-abstract-like figure (like figure 4) in the first section to show what and how they do in this paper. In this way, readers can easily understand the main idea of the paper.
4. The discussion is very meaningful. However, it seems that the authors simply stacked the content without specific organization rules. Please try your best to make them more organized and clear.
5. The authors can add some future research directions in Conclusions.
Overall, I think it is an interesting and meaningful work, and I can recommend it for publication after minor revisions.
Author Response
Comments 1: It would be better for the authors to give more detailed explanations about the used terminologies, such as FIFA 11+ program. These terms may be not familiar for readers.
Response 1: Thank you for this helpful suggestion. We appreciate the importance of making the terminology accessible to all readers. In response, we have added brief explanations and context for terms like the FIFA 11+ program in the relevant sections. The revision can be found on page 2, paragraph 6, lines 53–60, where we introduce the FIFA 11+ program with a clear description. We have also ensured that other specialized terms are defined or described to improve accessibility for all readers.
Comments 2: The organization is good. However, I prefer a shorter background and related work. Currently, Section 2 is too tedious.
Response 2: Thank you for your constructive comments on the organization of the paper. We have taken your advice into account and revised Section 2 (Background and Related Work) to make it more concise. Redundant information has been removed, particularly in pages 4–7, where we previously provided extensive background on related technologies. The literature review now focuses on the most relevant studies that directly inform our research, and the discussion of smart textiles and biomechanics has been condensed while retaining essential information. These revisions ensure that Section 2 remains informative yet succinct, without compromising the depth of context.
Comments 3: The authors are suggested to give a graphical-abstract-like figure (like figure 4) in the first section to show what and how they do in this paper. In this way, readers can easily understand the main idea of the paper.
Response 3: We appreciate your suggestion to include a graphical abstract-like figure in the first section. In response, we have added Figure 1 on page 3, which visually summarizes our research concept. This figure provides a clear representation of the smart socks-based biofeedback system, the process of real-time biomechanical monitoring, and the integration of mixed reality feedback. We believe this will help readers quickly grasp the core objectives and methodology of the study.
Comments 4: The discussion is very meaningful. However, it seems that the authors simply stacked the content without specific organization rules. Please try your best to make them more organized and clear.
Response 4: Thank you for your constructive feedback on the organization of the discussion. We understand the importance of a clear structure and have reorganized the content to improve its flow and readability. Revisions can be found in pages 19–22, where the discussion is now divided into structured subsections. These include an introduction to the DAid® system and its role in injury prevention, a comparison with existing injury prevention programs, a detailed examination of the biomechanical benefits of real-time feedback, a discussion of technological innovations such as the integration of smart textiles and mixed reality, and an exploration of challenges and future directions. These structural improvements make the discussion more logical and easier to follow, ensuring that each key topic receives appropriate attention.
Comments 5: The authors can add some future research directions in Conclusions.
Response 5: Thank you for your thoughtful suggestion. We agree that adding future research directions to the Conclusions (page 23, paragraph 2, lines 868–876) can provide useful context for ongoing work in this area. We have included discussions on expanding the system’s applications to other sports, refining personalized biofeedback to adapt to individual biomechanics, exploring remote monitoring through cloud-based platforms, and addressing technical challenges such as reducing latency and improving wireless data transmission. By incorporating these points, we provide a clearer vision of how the research can evolve and contribute to broader applications in sports science.
Round 2
Reviewer 2 Report
Comments and Suggestions for Authors
Authors made modifications to the manuscript according to the comments made in the previous revision. However, the revised manuscript provided did not highlight in any way the changes, which made the task of revising the new manuscript harder than usual.
No more comments.